# Statins to reduce renal sinus fat among breast cancer patients undergoing anthracycline-based chemotherapy: A substudy of PREVENT-WF-98213

William W. Hundley[1], Nathaniel S. O'Connell[2], Danielle L. Kirkman[3], Kristine C. Olson[1], Kerryn W. Reding[4], Bonnie Ky[5], Kathryn J. Ruddy[6], Glenn J. Lesser[7], Ralph B. D'AgostinoJr[2], W. Gregory Hundley[1,8]*, Moriah P. Bellissimo[1]*

1 Department of Internal Medicine, Division of Cardiology, Pauley Heart Center, Virginia Commonwealth University School of Medicine, Richmond, Virginia, United States of America, 2 Department of Biostatistics and Data Science, Wake Forest University School of Medicine, Winston-Salem, North Carolina, United States of America, 3 Department of Kinesiology and Health Sciences, Virginia Commonwealth University, Richmond, Virginia, United States of America, 4 Department of Biobehavioral Nursing and Health Informatics, University of Washington School of Nursing, Seattle, Washington, United States of America, 5 Department of Medicine, Perelman School of Medicine at the University of Pennsylvania, Philadelphia, Pennsylvania, United States of America, 6 Department of Oncology, Division of Medical Oncology, Mayo Clinic, Rochester, Minnesota , United States of America, 7 Department of Internal Medicine, Section on Hematology and Oncology, Wake Forest University School of Medicine, Winston-Salem, North Carolina, United States of America, 8 Departments of Internal Medicine and Radiology, Wake Forest University School of Medicine, Winston-Salem, North Carolina, United States of America

* greg.hundley@vcuhealth.org (WGH); Moriah.bellissimomyers@vcuhealth.org (MPB)

## Abstract

### Objective

To test if statin administration attenuated renal sinus fat (RSF) accumulation and if RSF was associated with renal function in women with breast cancer (BC) receiving anthracycline-based chemotherapy.

### Methods

This was a secondary analysis in a subgroup of women with stage I-III BC randomized to placebo (n = 35) or statin (40 mg/day atorvastatin, n = 44) therapy. At baseline and 24-months after randomization, RSF and intraabdominal fat were measured from magnetic resonance images, and estimated glomerular filtration rate (eGFR) was calculated from serum creatinine.

### Results

Participants in this study averaged 51 years of age (SD 11), 87% reported White race, and had a mean BMI (±SD) of 30.2 kg/m$^2$ (±6.1). Most participants (60%) were diagnosed with stage II BC. At 24-months, RSF was higher in the placebo

**Data availability statement:** The data that support the findings of this study are not publicly available because the data contain potentially identifying or sensitive patient information and the Institutional Review Board does not allow for public storage of the data. Databases for the study are stored with the Wake Forest NCI Community Oncology Research Program (NCORP). For inquiries regarding data access, please reach out the Wake Forest NCORP at NCORP@wakehealth.edu.

**Funding:** Funding provided by American Heart Association grant 23SFRNPCS1063854 (RBD, WGH, MPB), Susan G. Komen grant CTA241184399 (MPB), and National Institutes of Health grants R01HL118740 (RBD, WGH), K99HL173554 (MPB), T32CA093423 (MPB), UG1CA189824 (NSO, GJL, RBD), UG1CA189828 (BK), and UG1CA189823 (KJR). The funders did not play any role in the study design, data collection and analysis, decision to publish, or preparation of the manuscript.

**Competing interests:** The authors have declared that no competing interests exist.

group relative to the statin group (β [95% CI], p-value: 0.17 [0.009, 0.34], p = 0.04). After adjusting for baseline RSF, this signal remained but was attenuated (β [95% CI], p-value: 0.12 [−0.06, 0.29], p = 0.18). In all participants at baseline and prior to beginning chemotherapy for BC or study drug, higher RSF was associated with lower eGFR values in all participants (r = −0.23, p = 0.03). At 24-months by study group, greater RSF was associated with decreased eGFR in the placebo group (−0.51, p = 0.01) but not in the statin group (−0.25, p = 0.19).

## Discussion

Statin administration may lower RSF during anthracycline-based chemotherapy. These findings merit further investigation to determine if statins protect renal function during BC treatment.

## Introduction

Breast cancer (BC) is the most prevalent form of cancer affecting women in the United States. [1] Advances in BC detection and adjuvant treatment have contributed to a significant decline in BC-specific mortality rates and increase in long-term BC survivorship. However, some women experience adverse changes in body composition during cancer treatment, including increases in fat mass and declines in muscle quality. [2–4] In turn, these changes are linked to negative clinical outcomes, including increased all-cause and cardiovascular mortality risk and decreased cardiorespiratory fitness. [5–8] Therapeutic strategies are needed to reduce the adverse effects of adiposity and promote a healthy cancer survivorship.

Hydroxymethylglutaryl-CoA (HMG-CoA) reductase inhibitors (i.e., statins) are a widely used lipid-lowering agent that have been suggested as a therapeutic strategy to mitigate adverse chemotherapy effects [9–11] and reduce intraabdominal and epicardial fat. [12,13] Additionally, subcomponents of intraabdominal fat, such as increased amounts of renal sinus fat (RSF), have been associated with renal dysfunction and hypertension in non-cancer populations. [10,14,15] Yet, little is known regarding RSF and associations with renal function in women with BC. Importantly, impaired renal function increases the risk of all-cause and BC-specific mortality. [16] As BC survivorship continues to improve, there is a need to understand factors that influence the long-term health of these individuals and identify strategies to support health. This subgroup analysis of a randomized trial investigated if statin administration during receipt of chemotherapy was associated with changes in RSF among women treated for BC, and tested if RSF was associated with kidney function.

## Methods

### Study design and participants

Data for the current study were obtained courtesy of the Wake Forest National Cancer Institute Community Oncology Research Program (NCORP). The study was approved by the Wake Forest University School of Medicine Institutional Review

Board, and all participants provided written informed consent prior to enrollment. The parent study (PREVENT-WF-98213) was a double-blind, randomized, placebo controlled trial whose purpose was to determine if statins administered to individuals undergoing anthracycline-based chemotherapy would attenuate declines in left ventricular ejection fraction (LVEF) observed during treatment. [9] The primary study start date was February 2014, and the study was completed in September 2020. Data were collected over 24 months and participants were randomized 1:1 to receive a placebo or 40 mg of atorvastatin per day with both investigators and participants blinded to study group allocation. The study protocol and results were previously published. [9] Brief inclusion criteria were a diagnosis of stage I-III breast cancer or stage I-IV lymphoma, scheduled to receive adjuvant chemotherapy with an anthracycline, and at least 21 years of age. Brief exclusion criteria were use of a lipid-lowering agent within the last 6 months at screening, current postmenopausal hormone-replacement therapy, liver disease, untreated hypothyroidism, inflammatory conditions, or unstable angina, ventricular arrhythmias, or atrial fibrillation.

A total of 279 patients were enrolled in the parent trial with results previously published. [9] The present study included female BC participants only, so males (n = 23) and female patients with lymphoma (n = 19) were excluded. In addition, participants from racial groups including Native Hawaiian/Pacific Islander (n = 4), Native American/Alaskan (n = 2), Asian (n = 2) or unknown race (n = 1) represented <5% of the parent trial and were excluded in this study due to the inability to draw statistical inference from small groups. Thus, the present secondary analysis included a subset of women diagnosed with BC of White or Black race with renal sinus fat assessments at baseline (n = 79, Fig 1) or 24 months (n = 54). Supplemental table 1 compares demographic and clinical characteristics of the participants included in this study compared to the rest of the cohort.

## Renal sinus fat

Fat within the renal sinus as well as subcutaneous fat and intraabdominal fat (retroperitoneal and intraperitoneal depots) were measured from magnetic resonance images (MRI) collected at baseline and 24-months as previously described. [2] Single T1-weighted images were collected at the L2 vertebrae in the axial plane and analyzed at each visit using SliceO-matic software (Tomovision) by one technician blinded to all patient identifiers and study group assignment. [17] The L2 slice position was selected as it represents a position in the abdomen where the renal sinus is captured and intraabdominal and subcutaneous fat depots are representative of overall body composition. All images that visualized renal sinus fat

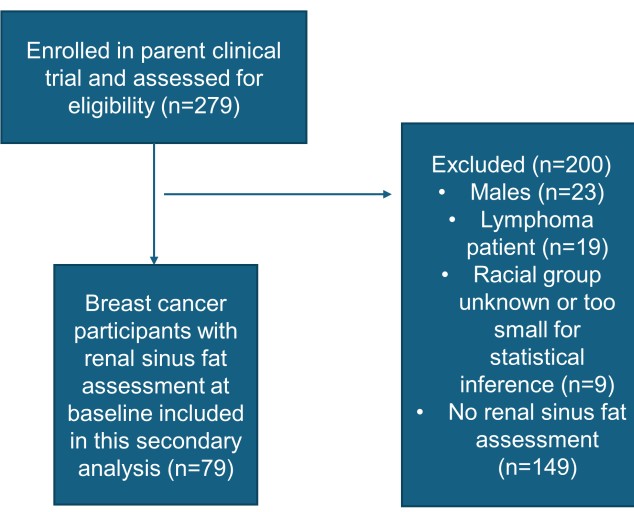

**Fig 1. Flow chart of included study participants.**

were analyzed, which included 79 participants at baseline and 54 participants at 24-months. Fig 2 displays a segmented axial image and the tagged depots.

## Renal function

Serum creatinine levels were available in 79 participants at baseline and 54 participants at 24 months. Estimated glomerular filtration rate (eGFR) was calculated using serum creatinine values following the race free CKD-EPI equation, which accounts for individual age and sex. [18] Individuals were also classified according to chronic kidney disease (CKD) stages by eGFR or normal kidney function (>90 mL/min/1.73m$^2$).

## Statistical analyses

Univariate analyses were performed for all continuous and categorical variables and are reported as mean±standard deviation (SD) or count and percent, respectively. Renal sinus fat values were square root-transformed to account for non-normal distributions. Chi-squared tests were used to compare proportions of the cohort by CKD stages, and t-tests were used to compare body composition and kidney function measures at baseline and 24 months. Linear regression analyses were used to assess if statin administration impacted RSF from baseline to 24-months by comparing RSF values at 24 months between placebo and statin groups. Thereafter, we performed a linear regression model analysis with 24-month RSF as the outcome and placebo or statin group as the main predictor while also adjusting for baseline RSF. To test if RSF was associated with kidney function at baseline (i.e., prior to beginning chemotherapy or allocation to study group), Pearson correlations were performed with RSF and eGFR. Partial correlations were then conducted to test if RSF was associated with renal function independent of intraabdominal fat or BMI. At 24-months, to explore differences in RSF and kidney function between study groups, correlations were performed stratified by placebo or statin group. Analyses were performed in JMP Pro (Version 16, SAS Inc, Cary, NC, USA).

## Results

Baseline demographic and clinical characteristics are shown in Table 1 for the whole cohort and by placebo and statin groups. Women in this cohort were middle-aged and 87% of participants reported White race. Of the 79 participants, 44 (56%) were randomized to receive the statin. On average, the women were obese according to BMI. About 20% were taking anti-hypertensive medications and over half were diagnosed with stage II BC. Characteristics of this subset of BC participants were similar to those excluded (S1 Table). Descriptive results for body composition and renal function at baseline to 24-months are shown in Table 2.

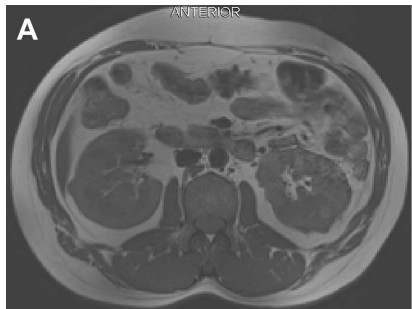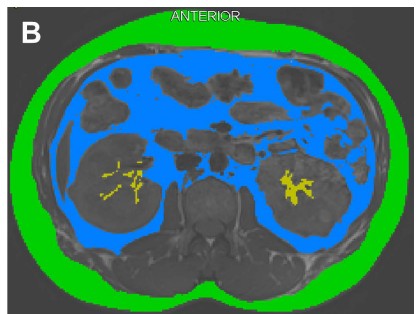

**Fig 2. Axial T1 weighted image taken at the L2 vertebrae. (A)** Black and white image. **(B)** Segmented imaging depicting renal sinus fat (gold), subcutaneous fat (green), and intra-abdominal fat (blue).

**Table 1. Baseline demographic and clinical characteristics of women with breast cancer.**

| Characteristic | Total cohort N=79 | Placebo n=35 | Statin n=44 |
|---|---|---|---|
| **Study group** | | | |
| Statin | 44 (56) | 0 (0) | 44 (100) |
| Age (years) | 51.3±10.9 | 52.8±10.8 | 50.0±10.9 |
| **Race** | | | |
| White | 69 (87) | 31 (89) | 38 (86) |
| Black | 10 (13) | 4 (11) | 6 (14) |
| **Body mass index (kg/m²)** | 30.2±6.1 | 30.5±6.5 | 29.9±5.9 |
| **Hypertension medications** | 32 (16) | 8 (23) | 8 (19) |
| **Cancer stage** | | | |
| I | 8 (10) | 5 (14) | 3 (7) |
| II | 47 (60) | 18 (51) | 29 (66) |
| III | 24 (30) | 12 (34) | 12 (27) |
| **Cumulative anthracycline dose (mg/m²)** | 236.5±26.7 | 236.9±37.8 | 236.3±12.3 |

Data are presented as mean±SD or n (%).

N missing for baseline hypertension medications=2

N missing for baseline cumulative anthracycline dose=3

**Table 2. Descriptive data for fat depots and renal function at baseline and 24 months in women with breast cancer.**

| Characteristic | Baseline | | 24 Months | |
|---|---|---|---|---|
| | Placebo (n=35) | Statin (n=44) | Placebo (n=25) | Statin (n=29) |
| **Renal sinus fat (cm²)** | 2.31±1.92 | 1.97±1.58 | 2.31±1.85 | 1.55±1.96 |
| **Intraabdominal fat (cm²)** | 123.3±65.1 | 109.4±53.7 | 132.6±59.8 | 123.7±59.7 |
| **Body mass index (kg/m²)** | 30.5±6.5 | 29.9±5.9 | 31.3±7.4 | 28.5±6.0 |
| **eGFR (ml/min/1.73m²)** | 91.4±17.6 | 93.7±14.7 | 92.2±18.0 | 86.6±17.2 |
| **Serum creatinine (mmol/L)** | 65.9±9.2 | 65.3±10.3 | 67.1±11.2 | 69.2±9.3 |

Data are presented as unadjusted mean±SD or n (%)

N=24 for intraabdominal fat at 24 months in the placebo group

Table 3 shows results testing for a difference in RSF at 24 months between the statin and placebo groups. At 24-months, the placebo group had higher levels of RSF compared to the statin group (β [95% CI], p-value: 0.17 [0.009, 0.34], p=0.04). In the second model adjusting for baseline levels of RSF, the placebo group tended to have a higher amount of RSF, but the signal was attenuated (β [95% CI], p-value: 0.12 [−0.06, 0.29], p=0.18). Baseline RSF was a strong predictor of 24-month levels of RSF (β [95% CI], p-value: 0.73 [0.46, 1.01], <0.001).

In all participants at baseline (n=79) prior to initiating BC treatment and study drug allocation, a greater amount of RSF was associated with worse eGFR (r=−0.23, p=0.03). This relationship remained when conducting partial correlations with BMI (r=−0.30, p<0.001) and intraabdominal fat (r=−0.24, p=0.002). At 24-months by study group, higher RSF was associated with lower eGFR (−0.51, p=0.01, n=25) in the placebo group but not associated with eGFR in the statin group (−0.25, p=0.19, n=29). An example participant randomized to the statin group who showed a decline in RSF area from baseline to 24 months is shown in Fig 3.

According to CKD stages, at baseline, 56% (n=44) of the cohort had normal kidney function (eGFR≥90 mL/min/1.73m²), 43% (n=34) met criteria for Stage G2 indicating mild loss of kidney function (eGFR 60–89 mL/min/1.73m²),

**Table 3. Analyses comparing renal sinus fat at 24-months between the placebo and statin groups.**

| Model 1 Outcome: 24-month renal sinus fat | | | |
|---|---|---|---|
| | β±standard error | 95% CI | p-value |
| Intercept | 1.25±0.08 | 1.08, 1.41 | <0.001 |
| Placebo group | 0.17±0.08 | 0.009, 0.34 | 0.04 |
| **Model 2 Outcome: 24-month renal sinus fat** | | | |
| | β±standard error | 95% CI | p-value |
| Intercept | 0.33±0.21 | −0.09, 0.76 | 0.12 |
| Placebo group | 0.12±0.08 | −0.06, 0.29 | 0.18 |
| Baseline renal sinus fat | 0.73±0.14 | 0.46, 1.01 | <0.001 |

Renal sinus fat values were square root transformed to account for a non-normal distribution. The first model shows results from a Student's t-test comparing 24-month renal sinus fat between groups (n = 54). The second model shows results from a linear regression model comparing 24-month renal sinus fat between groups while adjusting for baseline renal sinus fat (n = 35).

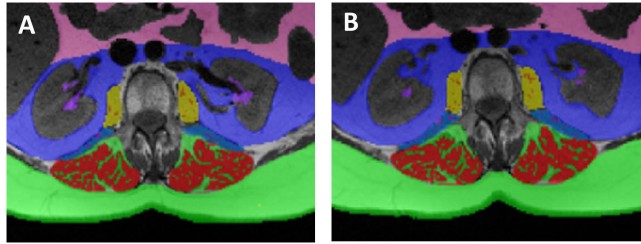

**Fig 3. Tagged axial T1 weighted image taken at the L2 vertebrae showing an example of a participant who received a statin and had a decline in renal sinus fat (purple tagging) from baseline (A, 2.64 cm$^2$) to 24-months (B, 0.71 cm$^2$).**

and 1% (n = 1) met criteria for Stage G3 indicating mild to moderate kidney function loss (eGFR range 30–59 mL/min/1.73m$^2$; minimum eGFR value in cohort = 47 mL/min/1.73 m$^2$). At 24 months, 48% (n = 26) of participants had normal renal function, 43% (n = 23) met criteria for Stage G2, and 9% (n = 5) met criteria for Stage G3 (p = 0.08).

When RSF levels were compared by CKD stages, individuals with normal renal function (eGFR > 90 mL/min/1.73m$^2$) had lower levels of RSF compared to individuals whose eGFR values were <90 mL/min/1.73m$^2$ at baseline (p = 0.06) and 24 months (p = 0.099, Fig 4). Levels of inflammatory biomarkers were not associated with amount of RSF (S2 Table).

## Discussion

In a subgroup of women with stage I-III BC undergoing anthracycline-based chemotherapy and randomized to 40 mg/day atorvastatin or placebo, individuals in the statin group tended to exhibit a decrease in RSF relative to those in the placebo group at 24 months. In addition, increased RSF was associated with a lower eGFR at baseline in all participants prior to beginning chemotherapy treatment and study group allocation. This association remained at 24 months in the placebo group, but the statin group did not exhibit a statistically significant association between RSF and eGFR. These results warrant further investigation to confirm if statin administration may reduce RSF, a risk factor for kidney disease and hypertension. [17,19]

As BC survivorship continues to increase, strategies are needed to improve the long-term health of these individuals. Preventing adverse changes in body composition during treatment may be a key factor that can support a healthy BC survivorship. Some cancer treatments can be damaging to the kidneys, and while preclinical work and limited human studies show that RSF is associated with worse kidney function, little evidence regarding RSF is available in BC populations.

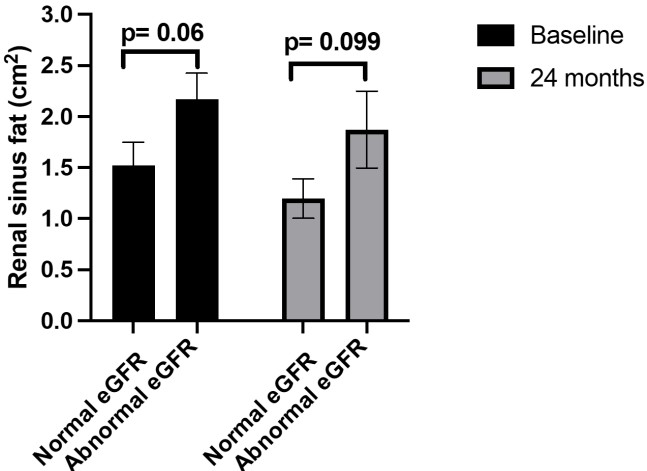

**Fig 4. Renal sinus fat levels according to eGFR groups at baseline and 24 months.** Normal eGFR values were considered ≥90 mL/min/1.73m², in line with chronic kidney disease stage 1 criteria. Abnormal eGFR was considered as <90 mL/min/1.73m², indicating some loss of kidney function following chronic kidney disease stages 2-3.

Here, findings suggest there may be evidence of an improvement in RSF during chemotherapy with the administration of a statin. This finding was marginally significant comparing groups at 24-months. After accounting for baseline RSF, this finding was attenuated but the results were still suggestive of an improvement in RSF by statin therapy in a small group of participants (n = 35). As the amount of RSF in the placebo group did not change, this may suggest that anthracycline-based chemotherapy does not worsen RSF levels but statin administration may help reduce RSF. Amounts of RSF in this study were lower than the average reported in the Framingham cohort (0.97 cm², 25th percentile: 0.73, 75th percentile: 1.34) but similar to the median reported in obese participants prior to bariatric surgery (2.3 cm² [interquartile range: 1.7–3.1]). [15,19]

In non-cancer populations, greater RSF is associated with renal dysfunction, but this relationship is unknown in women with BC. In this cohort, we found higher amounts of RSF were associated with lower eGFR values in all participants at baseline. At 24 months, this association persisted within the placebo group, but not in the statin group, suggesting a potential benefit of the statin. As RSF is increased in patients with obesity, we performed partial correlations adjusted for BMI and intraabdominal fat, and found that greater RSF remained independently correlated with lower eGFR values. Moreover, participants with normal renal function (eGFR > 90 mL/min/1.73 m²) tended to have lower amounts of RSF relative to participants with impaired renal function (eGFR < 90 mL/min/1.73 m², Fig 4). While the sample size in this study is small, previously published work supports these results. In non-cancer populations, increased RSF also correlated with lower eGFR values in individuals with obesity and type 2 diabetes. [14,15] Participants in the Framingham Heart Study with renal sinus fat above sex-specific >90th percentiles (renal sinus fat ≥ 0.445 cm² in women and ≥0.71 cm² in men) had increased risk for CKD independent of BMI (odds ratio: 1.86, p = 0.04) and visceral fat (odds ratio: 1.86, p = 0.05). [19] Findings reported here corroborate others in non-cancer populations showing increased RSF as a factor associated with lower eGFR and a possible benefit of statin administration to reduce RSF and protect eGFR.

Accumulation of RSF may alter renal function through structural and functional changes. Increased presence of RSF can constrict the renal vein and lead to greater kidney volume and intrarenal pressure, and altered sodium excretion. [20] Much of this work is based on animal models. Intraabdominal pressure may also be further amplified in individuals with abdominal obesity. [20] In addition, compression of renal structures may lead to activation of the renin-angiotensin-aldosterone system, which can promote diseases such as hypertension, insulin resistance, and atherosclerosis. [20,21]

Finally, other ectopic fat depots are highly lipolytic and pro-inflammatory, [22] which may contribute to injury of the renal glomeruli. [23] Further work is needed to understand these mechanisms and if they differ cancer and non-cancer populations.

Mechanisms for how statins may reduce RSF are unknown. However, in other studies that showed improvements in epicardial fat and the ratio of visceral fat to abdominal subcutaneous fat ratio with statin administration, it is postulated that activation of peroxisome proliferator-activated receptors (PPAR) by statins increases fatty acid oxidation to reduce these fat depots. [12,24] An additional concept may be that increased inflammation during anthracycline-based chemotherapy treatment could lend to increased ectopic fat deposition, [22] but the anti-inflammatory effects of statins could offer protection from ectopic fat deposition. [25] In the parent trial, the inflammatory markers c-reactive protein, interleukin-6, and tumor necrosis factor-α were not different in the placebo and statin groups. [9] In this subset of the parent study, those markers of inflammation were not associated with RSF. Further work in a larger cohort is needed to confirm if statin administration reduces RSF and mechanisms by which this occurs.

eGFR values did not change significantly over 24-months of BC treatment in the placebo or statin groups. Changes in eGFR and serum creatinine from baseline to 24-months were within the normal 30% daily variation that can be observed, and thus these changes are not likely to reflect clinically relevant declines in renal function. [26] Other common cancer treatments such as cisplatin may be more damaging to the kidneys than anthracyclines. [27]

By CKD stages, a trend was noted for the change in proportion of participants meeting criteria for abnormal kidney function. At baseline, 44% of participants had abnormal kidney function compared to 52% at 24-months. These analyses were descriptive and not adjusted for confounding variables although eGFR calculations account for age and sex. In line with our findings, previous reports have described a 53–57% prevalence of renal impairment in patients with advanced cancer. [28] Cancer treatment may adversely impact particular individuals. For example, individuals with advanced cancer or elderly individuals may be at greater risk for renal toxicity. [29] Indeed, monitoring of renal function is recommended for individuals treated for cancer. [28] The presence of additional comorbidities may also impact a person's risk for renal function decline. While we were not able to explore these subgroups here due to a small sample size, future studies in larger cohorts could examine the impacts of cancer treatment on renal function and if statin use impacts those findings.

Statin dosing can vary and the timing of statin administration may also impact findings. In a meta-analysis of randomized controlled trials including >110,000 participants, statin administration slowed eGFR decline; however, this meta-analysis was not conducted in cancer patients, and statin dosage ranged from 10–80 mg of various statins. [30] Here, all participants in the statin group received 40 mg/day of atorvastatin, and participants began statin or placebo therapy 24–48 hours before beginning chemotherapy. [9] Therefore, it cannot be determined from this study if initiating statins earlier, for a longer time, or at a different dose would alter renal function or RSF in a different manner than is reported here.

Strengths of this study include the prospective design and randomization to study groups to limit bias and differences between groups. Additionally, MRI is considered a gold-standard for assessment of ectopic fat depots. While renal toxicity is reported in populations with cancer, there is little data available on renal function or RSF for individuals going through BC treatment, and this work adds to that literature. There are also several limitations to this study. The majority of the cohort reported White rice, and Black race was the only other race included due to small group sizes. Diverse patient populations may respond differently to treatments, and the limited racial diversity of this study is not reflective of the general population of patients with BC, which limits generalizability of the results. This study followed patients for 24-months; however, longer study duration may yield different results and a better understanding of the long-term impact of statins on RSF and renal health. The sample size limited our statistical power. With the mean difference detected here (0.23 units) we would require 95 participants per group to have 80% and a 5% level of significance. So, while we expect the mean difference reported here to be true, we were under powered to detect that difference statistically. We utilized data available to us in this cohort to explore a research question of clinical significance, and findings may warrant additional follow up. Finally, eGFR was calculated from one creatinine sample at each time point, which can be affected by dietary protein

intake, exercise, and/or lean mass. Future investigations should aim to reproduce the findings with repeated serum creatinine and/or serum cystatin c sampling.

In summary, the results of this secondary analysis in a subgroup of women receiving anthracycline-based chemotherapy for the treatment of stage I-III BC and randomized to 40 mg/day of atorvastatin versus placebo over 24 months indicated that RSF tended to decrease in the statin group. Additionally, higher levels of RSF were associated with a lower eGFR in all participants at baseline and at 24-months among participants randomized to the placebo group. In larger cohorts, future research could provide valuable insights into the efficacy of statins being used as a pharmacological intervention to reduce RSF and understand the impact on renal health among cancer patients.

## Supporting information

**S1 Table. Supplemental Table 1. Comparison of baseline demographic and clinical characteristics of women with breast cancer in the parent study and current study.**
(DOC)

**S2 Table. Supplemental Table 2. Regression models of inflammatory markers (predictor) and renal sinus fat (outcome).**
(DOC)

**S1 File. Supplemental CONSORT Diagram.** CONSORT 2010 checklist of information to include when reporting a randomised trial.
(DOC)

## Acknowledgments

This was a secondary analysis of NCT01988571, and data were shared/used with permission from the principal investigator (WG Hundley). This secondary analysis included a subset of the parent study in which only female breast cancer participants were enrolled. We thank all study participants and staff for their efforts to make this study possible.

## Author contributions

**Conceptualization:** William W. Hundley, W. Gregory Hundley, Moriah P. Bellissimo.

**Formal analysis:** William W. Hundley, Nathaniel S. O'Connell, Ralph B. D'Agostino, Jr., W. Gregory Hundley, Moriah P. Bellissimo.

**Investigation:** W. Gregory Hundley.

**Methodology:** Nathaniel S. O'Connell, Danielle L. Kirkman, Kerryn W. Reding, Bonnie Ky, Kathryn J. Ruddy, Glenn J. Lesser, Ralph B. D'Agostino, Jr., W. Gregory Hundley, Moriah P. Bellissimo.

**Writing – original draft:** William W. Hundley, W. Gregory Hundley, Moriah P. Bellissimo.

**Writing – review & editing:** William W. Hundley, Nathaniel S. O'Connell, Danielle L. Kirkman, Kristine C. Olson, Kerryn W. Reding, Bonnie Ky, Kathryn J. Ruddy, Glenn J. Lesser, Ralph B. D'Agostino, Jr., W. Gregory Hundley, Moriah P. Bellissimo.

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
