## [Decision Letter · Decision Letter 0]

3 Mar 2025

Dear Dr. Bellissimo,

Thank you for submitting your manuscript to PLOS ONE. After careful consideration, we feel that it has merit but does not fully meet PLOS ONE’s publication criteria as it currently stands. Therefore, we invite you to submit a revised version of the manuscript that addresses the points raised during the review process.

We look forward to receiving your revised manuscript.

Kind regards,

Amir Hossein Behnoush

Academic Editor

PLOS ONE

Journal Requirements:

5. Please include a copy of Table 3 which you refer to in your text on page 10.

Reviewers' comments:

Reviewer's Responses to Questions

**Comments to the Author**

1. Is the manuscript technically sound, and do the data support the conclusions?

Reviewer #1: Yes

Reviewer #2: Yes

Reviewer #3: Yes

2. Has the statistical analysis been performed appropriately and rigorously?

Reviewer #1: No

Reviewer #2: Yes

Reviewer #3: Yes

3. Have the authors made all data underlying the findings in their manuscript fully available?

Reviewer #1: Yes

Reviewer #2: Yes

Reviewer #3: Yes

4. Is the manuscript presented in an intelligible fashion and written in standard English?

Reviewer #1: Yes

Reviewer #2: Yes

Reviewer #3: Yes

Reviewer #1: Recommedation

Major Revision -

The paper does not make an easy read.

As I understand it from lines 92-97, this is an analysis of a subgroup of 79 breast cancer (surviving?) patients of White or Black race selected from the larger Prevent randomised trial which included 279 patients. This (subgroup) fact needs more prominence within the title of the paper and in the Abstract.

In such an example, Placebo v Statin is not a covariate but is the main design variable. In such a situation the simple comparison of, for example, fat levels at 24 months between the two groups should form the primary analysis and take the form Model I: Fat24 = α + βTreat. Secondary to that, and arguably the better, analysis would use a linear regression model Model II: Fat24 = α + βTreat + γFat0. If the second β differs from that of the first model then model II should be used for reporting. If it is very similar, then the (simpler) Model I provides the better summary. If (say) Model I is selected, the covariates one by one can then be added and only if they individually seriously affect β should they be retained in the model. Another option is to calculate the paired differences d = Far24 – Fat0 and then compare the corresponding difference of the means of the Placebo and Statin groups using a z-test. However, as this analysis uses a ‘difference of differences’ it is not so easy to interpret.

However, a major problem with this paper is that confidence intervals of the β’s quoted are not given.

Since Placebo and Statin are the main design variables, they should be the first row of Table 1.

In general, it is of no value to make statistical comparisons of the covariates (say between age groups) as their effect on any difference between the two design groups is taken account of in the modelling process. So, the p-values in Table 1 should be omitted. Listing the characteristics by design group is nevertheless important. Further the r2 values provide no useful information and can also be omitted.

Line 143. Setting a fixed value for significance is not a good idea neither is it likely that the modelling approach with an Interaction term included is justified.

Reviewer #2: The authors have conducted an important study looking into the use of Statins to reduce renal sinus fat among breast cancer patients undergoing anthracycline-based chemotherapy. They have highlighted a lot of useful findings with regards to statin administration and the effects on renal sinus fat in these patients. They have also highlighted the limitations to the study as well as the directions future studies could take.

Reviewer #3: Dear authors,

Congratulations on a well-structured, informative, and well-written manuscript. The study is clinically relevant, and the presentation of methods, results, and discussion is clear and logically organized. The use of MRI-based renal sinus fat quantification and the exploration of its relationship with renal function in breast cancer patients undergoing chemotherapy provide valuable insights. The manuscript effectively highlights an important area, and the discussion thoughtfully integrates findings with existing literature. However, I have a minor comment which you may address:

1- If the renal function outcomes (eGFR, CKD stage progression) and MRI-based RSF measurements were not formally adjudicated, the authors should state this as a limitation in the discussion.

2- Also Please clarify how allocation concealment was implemented before randomization.

**Do you want your identity to be public for this peer review?** For information about this choice, including consent withdrawal, please see our Privacy Policy

Reviewer #1: No

Reviewer #2: No

Reviewer #3: **Yes: ** Sanam Alilou

---

## [Author Response · Author response to Decision Letter 1]

16 Apr 2025

Hello,

As instructed per our decision letter, we have uploaded our Response to Reviewers as a separate file.

Thank you!

---

## [Decision Letter · Decision Letter 1]

15 May 2025

Dear Dr. Bellissimo,

We look forward to receiving your revised manuscript.

Kind regards,

Amir Hossein Behnoush

Academic Editor

PLOS ONE

Reviewers' comments:

Reviewer's Responses to Questions

**Comments to the Author**

Reviewer #1: (No Response)

Reviewer #2: All comments have been addressed

Reviewer #4: (No Response)

2. Is the manuscript technically sound, and do the data support the conclusions?

Reviewer #1: Yes

Reviewer #2: (No Response)

Reviewer #4: Partly

3. Has the statistical analysis been performed appropriately and rigorously?

Reviewer #1: No

Reviewer #2: (No Response)

Reviewer #4: No

4. Have the authors made all data underlying the findings in their manuscript fully available?

Reviewer #1: Yes

Reviewer #2: (No Response)

Reviewer #4: No

5. Is the manuscript presented in an intelligible fashion and written in standard English?

Reviewer #1: No

Reviewer #2: (No Response)

Reviewer #4: Yes

Reviewer #1: Major Revision

This is a very disappointing revision that takes little note of the suggestions I made in my earlier review.

Focussing on the Results section of the current Abstract, it begins by presenting differences in patient characteristics in the two groups. The real focus should be on Statin v Placebo and if a difference between these groups is demonstrated, is the size of the effect influenced by the covariates?

My earlier review emphasised the need for confidence intervals. The authors have calculated some, but none of these are included in the Abstract. Thus the Abstract needs redrafting with a clear focus on Statin v Placebo with respect to RSF.

The Discussion section is very weak indeed.

Some of other points of my first review are repeated below:

This (subgroup) fact needs more prominence within the title of the paper and in the Abstract.

Further the r2 values provide no useful information and can also be omitted.

In general, it is of no value to make statistical comparisons of the covariates (say between age groups) as their effect on any difference between the two design groups is taken account of in the modelling process.

Setting a fixed value for significance is not a good idea neither is it likely that the modelling approach with an Interaction term included is justified.

Reviewer #2: (No Response)

Reviewer #4: Methods/Line 146-148: “Additional covariates considered but not included as they were not associated with renal sinus fat were cancer stage, education level, marital status, income, smoking status, and cumulative anthracycline dose.”

Please indicate why you have not applied a multivariate analysis, since your study group is relatively small (~79 patients) and you need to assess parameters that associate with breast cancer, and HMG-CoA reductase inhibitors. You are including these to avoid confounding effects or biases due to small study size since your study size is not enough to provide sufficient randomization. Please also specify if you have conducted a linear/log/Poisson regression or cox survival analysis.

Methods/ Line 154 : please reason why 10% statistical significance was chosen. Please also cite suitable references in the same field that mention they have used the same values.

Methods: please provide sample size calculations, and cite relevant articles.

Tables: please change ß to OR, RR, or HR depending on your analysis. Please name the exact analysis that you have used. Please also justify why you have reported serum creatinine along with eGFR, and of what excess value does serum creatinine provides.

**Do you want your identity to be public for this peer review?** For information about this choice, including consent withdrawal, please see our Privacy Policy

Reviewer #1: No

Reviewer #2: No

Reviewer #4: **Yes: ** Alireza Ramandi

---

## [Author Response · Author response to Decision Letter 2]

21 Jul 2025

We have included our response to reviewer comments in our cover letter with this revision.

---

## [Decision Letter · Decision Letter 2]

29 Aug 2025

Statins to reduce renal sinus fat among breast cancer patients undergoing anthracycline-based chemotherapy: A substudy of PREVENT-WF-98213

PONE-D-24-56485R2

Dear Dr. Bellissimo,

We’re pleased to inform you that your manuscript has been judged scientifically suitable for publication and will be formally accepted for publication once it meets all outstanding technical requirements.

Kind regards,

Amir Hossein Behnoush

Academic Editor

PLOS ONE

Additional Editor Comments (optional):

Reviewers' comments:

Reviewer's Responses to Questions

**Comments to the Author**

Reviewer #1: All comments have been addressed

2. Is the manuscript technically sound, and do the data support the conclusions?

Reviewer #1: Partly

3. Has the statistical analysis been performed appropriately and rigorously?

Reviewer #1: Yes

4. Have the authors made all data underlying the findings in their manuscript fully available?

Reviewer #1: Yes

5. Is the manuscript presented in an intelligible fashion and written in standard English?

Reviewer #1: No

Reviewer #1: Minor revision but important

The authors have used the Greek beta (β) to describe the comparison between the two treatment groups without explanation of what it represents. A clear definition is required - particularly for the Abstract.

**Do you want your identity to be public for this peer review?** For information about this choice, including consent withdrawal, please see our Privacy Policy

Reviewer #1: No

---

## [Editor Report · Acceptance letter]

PONE-D-24-56485R2

PLOS ONE

Dear Dr. Bellissimo,

I'm pleased to inform you that your manuscript has been deemed suitable for publication in PLOS ONE. Congratulations! Your manuscript is now being handed over to our production team.

Kind regards,

on behalf of

Dr. Amir Hossein Behnoush

Academic Editor

PLOS ONE